# Determination of Ultra-Trace Cobalt in Water Samples Using Dispersive Liquid-Liquid Microextraction Followed by Graphite Furnace Atomic Absorption Spectrometry

**DOI:** 10.3390/molecules27092694

**Published:** 2022-04-22

**Authors:** Quan Han, Yaqi Liu, Yanyan Huo, Dan Li, Xiaohui Yang

**Affiliations:** 1School of Chemical Engineering, Xi’an University, Xi’an 710065, China; huoyeye@163.com (Y.H.); lidan5719@163.com (D.L.); sheepyangxh@163.com (X.Y.); 2School of Chemistry and Chemical Engineering, Yan′an University, Xi’an 716061, China; 16426958@163.com

**Keywords:** dispersive liquid–liquid microextraction, graphite furnace atomic absorptionspectrometry, cobalt, 2-(5-bromopyridyazo)-5-dimethylaminoanline, 1,2-dichloroethan, acetonitrile, water samples

## Abstract

A novel method for the determination of ultra-trace cobalt by dispersive liquid–liquid microextraction (DLLME) coupled with graphite furnace atomic absorption spectrometry has been developed. It is based on the color reaction of Co^2+^ with 2-(5-bromo-2-pyridylazo)-5-dimethylaminoaniline (5-Br-PADMA) in a Britton–Robinson buffer solution at pH 6.0 to form stable hydrophobic chelates, which were separated and enriched by DLLME with 1,2-dichloroethane (CH_2_ClCH_2_Cl) as extraction and acetonitrile (CH_3_CN) as a dispersive solvent. The sedimented phase containing the chelates is then determined with GFAAS. Parameters that affect extraction efficiency, such as types and volumes of extraction and disperser solvents, pH of sample solution, extraction time, concentration of the chelating agent 5-Br-PADMA, and salt effect, were investigated. Under optimal conditions, the calibration graph was linear over the range 0.05–1.0 ng/mL, with a correlation coefficient of 0.9922 and a detection limit of 0.03 ng/mL. Preconcentration factor (PF) is calculated as the ratio of the aqueous solution volume (5 mL) to that of the organic phase volume (40 μL), and enrichment factor (EF) is calculated as the ratio of the slopes of the calibration graphs obtained with and without DLLME for 5.0 mL of sample solution, which were 120 and 112.5, respectively. The extraction efficiency, calculated by EF/PF·100, was 93.8%. The relative standard deviation (RSD) at the 0.5 ng/mL Co^2+^ level was 3.8% (*n* = 6). The method has been applied to the determination of trace cobalt in water samples with satisfactory results.

## 1. Introduction

Cobalt is widely used in various industries as alloys, catalysts, batteries, paints, drugs and ceramics, but at the same time, it is a significant pollutant of the environment because of its toxic effect on human health. Cobalt is a typical metal ion present in biological and environmental samples and has important roles in many physiological functions. However, at high levels, it can be toxic and can lead to toxic effects such as vasodilatation, cardiomyopathy, low blood pressure, and bone defects in humans and animals [1,2]. Therefore, the development of efficient, reliable determination methods for monitoring the level of cobalt concentration in natural waters is required. However, the determination of trace cobalt in biological and environmental samples is very difficult due to its extremely low concentration and the interfering effects of the matrix. A highly sensitive analytical technique coupled with a separation and preconcentration approach is one of the best ways to solve these problems.

There are several methods reported for the determination of cobalt at low concentrations, including adsorptive stripping voltammetry (AdSV) [3,4,5], thermal lens spectrometry (TLS) [6,7], atomic fluorescence spectrometry (AFS) [8], graphite furnace atomic absorption spectrometry (GFAAS) [9,10,11], inductively coupled plasma atomic emission spectrometry (ICP-AES) [12,13,14], and inductively coupled plasma mass spectrometry (ICP-MS) [15,16,17]. GFAAS is selected here because it is a well-established and cost-effective technique and has the advantages of excellent sensitivity and small volumes of sample required.

Several methods have been reported for the separation and preconcentration of cobalt. Liquid–liquid extraction (LLE) is one of the most traditional and earliest extraction and pre-concentration techniques, but this method generally uses large organic solvents and involves a tedious procedure [18,19]. Solid-phase extraction (SPE), an alternative technique, requires much less organic solvent compared to LLE, but it is relatively expensive, and its enrichment factor is usually low [20,21,22]. Newly developed techniques include solid-phase microextraction [23], liquid-phase microextraction [24], single-drop microextraction [25], cloud-point extraction [6,7,26], and dispersive liquid–liquid microextraction (DLLME) [27,28]. The developments, advantages, and limitations of these newer methodologies have been discussed in some previous comprehensive reviews [29].

DLLME, a powerful preconcentration technique developed by Assadi and his co-workers in 2006 [30], is based on a binary mixture of solvents containing a high-density solvent with a very low water solubility as extractant, and a water-miscible one as the disperser to extract the objective compounds from an aqueous sample solution. The main advantages of DLLME are simplicity, fast extraction speed, low organic solvent consumption, low cost, and high recovery and enrichment factors. Furthermore, DLLME can easily be adapted for GFAAS and can improve the detection limit and selectivity of determination because of its low volume extracts.

The aim of this work is to develop a new method by combining DLLME with GFAAS for the determination of ultra-trace cobalt in environmental samples. The combination is favorable because DLLME is a highly efficient separation approach and involves preconcentration in a very small volume of solvent, and GFAAS is a highly sensitive detection technique and only requires a few microliters of sample to carry out the determination. In the developed system, a laboratory-synthesized reagent 2-(5-bromo-2-pyridylazo)-5-dimethylaminoaniline (5-Br-PADMA) was selected as the chelating agent, for it was proved to be a selective chromogenic reagent for the spectrophotometric determination of cobalt [31]. The structures of the reagent and its cobalt(II) complex are shown in Figure 1. The experimental parameters influencing the DLLME efficiency and GFAAS determination were investigated in detail, and optimum conditions were selected. The proposed method was successfully applied to the determination of ultra-trace amounts of cobalt in water samples.

## 2. Experimental Section

### 2.1. Apparatus

A model PinAAcle 900 T atomic absorption spectrophotometer (Perkin Elmer, Waltham, MA, USA) with a graphite furnace atomizer and Zeeman effects background correction was used for measurements. A model AS-2 cobalt hollow-cathode lamp (Beijing General Research Institute for Nonferrous Metals, Beijing, China), operated at a current of 13 mA and a wavelength of 240.7 nm with a spectral bandwidth of 0.2 nm, was employed. THGA graphite tubes (Perkin Elmer) were also employed. Argon of 99.99% purity (Xi’an Tenglong Chemical Co., Ltd., Xi’an, China) was used as a purge and protective gas. All measurements were based on the peak area. The optimum operating parameters for GFAAS are listed in Table 1. The pH values were measured with a model PHS-3C-01 pH lab meter furnished with a combined glass electrode (Shanghai San-Xin Instrumentation Inc., Shanghai, China). A model IKA Vortex-3 (IKA Works, Guangzhou, China) was used for mixing the reagents during extraction operation. Phase separation was performed using a model TD4A centrifuge (Hunan Kaida Scientific Instruments Co., Ltd., Changsha, China). A Milli-Q water purification system (Millipore Corporation, Billerica, MA, USA) was employed to prepare ultrapure water (≤0.055 μS/cm).

### 2.2. Synthesis of 5-Br-PADAM

The preparation of 5-Br-PADAM was described in detail in Ref. [31]. Dissolve 4.5 g (0.02 mol) of diazotate of 5-Br-2-amino-pryridine in 50 mL of aqueous ethanol solution (1:1), and cool the solution to 0 °C in an ice-salt bath. Dissolve 4.2 g (0.02 mol) of *N*,*N*-dimethylaniline hydrochloride in 50 mL of aqueous ethanol solution (1:1), and cool the solution to 0 °C. Add this solution to the diazotized solution. Add 6 mL of hydrochloric acid solution (1:1) dropwise to the mixture with vigorous stirring. Stir for 2–3 h, continue stirring for 4 h at room temperature, and then let it stand overnight. Adjust the pH value of the solution to 5–6 with 50% sodium acetate, filter off the precipitate, and wash with water. 

The filter cake was recrystallized from an ethanol–water solution (3:1), and a metallic dark red crystalline product was obtained.

### 2.3. Reagents and Solutions

Stock standard solution (1.0 mg/mL) of Co was obtained from the Beijing Tan-Mo Quality Testing Technology Co., Ltd. (Beijing, China). Working solutions were prepared by a stepwise dilution of the stock standard solution. A 5 × 10^−4^ mol/L 5-Br-PADAM (laboratory-synthesized [31]) solution was prepared by dissolving 0.0385 g of 5-Br-PADAM in 250 mL of ethanol. Britton–Robinson buffer solutions (pH 4–9) were prepared by adding 0.02 mol/L NaOH of solution to a mixture of 0.04 mol/L phosphoric acid, boric acid, and acetic acid solution to the required pH on the pH meter. Carbon tetrachloride (CCl_4_), 1,2-dichloroethane (C_2_H_4_Cl_2_), tetrachloroethene (C_2_Cl_4_), bromobenzene (C_6_H_5_Br), and 1,2-dichlorobenzene (C_6_H_4_Cl_2_) were purchased from Sinopharm Chemical Co. Ltd. (Shanghai, China). All chemicals used in this research were of analytical reagent grade or better, unless otherwise stated. Ultrapure water was used throughout the experimental work.

### 2.4. General Procedure

For the formation of cobalt chelate, an aliquot of the 5 mL of the standard or sample solution containing cobalt, 1.0 mL pH 6.0 Britton–Robinson buffer solution, and 120 μL of 5.0 × 10^−4^ mol/L 5-Br-PADAM ethanol solution were placed in a 10 mL screw cap glass test tube with a conical bottom. The mixture was shaken well and placed at room temperature for 5 min. Afterwards, for DLLME, 40 μL of 1,2-dichloroethane as an extraction solvent and 500 μL of acetonitrile as a dispersive solvent were rapidly injected into the sample solution by syringe, and the resultant solution was completely mixed by shaking for 2 min using a vortex shaker (speed scale 5), so that a cloudy solution was formed in the test tube. The mixture was then centrifuged for 5 min at 4000 rpm. After this process, the dispersed fine droplets of 1,2-dichloroethane were sedimented at the bottom of the test tube. Finally, using a microsyringe, 20 μL of the sediment phase at the bottom of the test tube was injected into the graphite tube, and the cobalt content was determined by graphite furnace atomic absorption spectrometry.

## 3. Results and Discussion

To achieve the highest extraction efficiency, the influence of varying different parameters, including types of extraction and disperser solvents and their volumes, the pH of the sample solution, extraction time, the concentration of the chelating agent 5-Br-PADMA, and the salt effect, were optimized. The enhancement factor (EF) was calculated from the slope ratio of calibration graphs obtained after and before DLLME.

### 3.1. Influence of Type and Volume of Extraction Solvent

The choice of an appropriate extraction solvent is of great significance for achieving good extraction efficiency, high enrichment factor, and selectivity for the analyte in the DLLME method. The extraction solvents should possess a higher density than water, as well as low water solubilities, and should have a high extraction efficiency for the Co(II)-5-Br-PADMA chelate from aqueous solution. Based on these facts, a variety of water-immiscible organic solvents such as carbon tetrachloride (CCl_4_), 1,2-dichloroethane (C_2_H_4_Cl_2_), tetrachloroethene (C_2_Cl_4_), bromobenzene (C_6_H_5_Br), and 1,2-dichlorobenzene (C_6_H_4_Cl_2_), with respective densities of 1.595, 1.26, 1.63, 1.50, and 1.30 g/cm^3^, were investigated as possible extraction solvents for the extraction of Co(II)-5-Br-PADMA chelate. The experimental conditions were fixed and performed using 500 μL of dispersive solvent and 40 μL of extraction solvent. Figure 1 shows the effect of the extraction solvent type on the extraction efficiency of the analytes. As it can be seen, the extraction efficiency of C_2_H_4_Cl_2_ is the highest. Thus, C_2_H_4_Cl_2_ was selected as the best extraction solvent for further experiments.

In order to evaluate the effect of extraction solvent volume, different volumes of C_2_H_4_Cl_2_ were examined with the same DLLME procedures. As seen in Figure 2, the analytical signal was increased up to 40 μL of C_2_H_4_Cl_2_ and then decreased significantly at larger volumes of the extraction solvent because of the increase in the volume of sediment phase, which, in turn, resulted in a decreased enrichment factor of the system. Therefore, 40 μL of C_2_H_4_Cl_2_ was selected as the optimum extraction solvent volume for subsequent experiments.

### 3.2. Influence of Nature and Volume of Disperser Solvent

The miscibility of the disperser solvent in organic phase (extraction solvent) and aqueous phase (sample solution) is the main point for the selection of the disperser solvent in DLLME. Several organic solvents such as acetone, acetonitrile, methanol, and ethanol, which show this property, were chosen as disperser solvents. A series of sample solutions were studied by using 500 μL of each disperser solvent and 40 μL of C_2_H_4_Cl_2_ (extraction solvent). It was observed that the volume of the sedimented phase was increased in the following order: methanol, ethanol, acetone, and acetonitrile. Furthermore, acetonitrile could give the highest signal response of the target analyte (Figure 3). So, acetonitrile was selected as the disperser solvent in the following experiment.

The influence of the volume of disperser solvent acetonitrile on the absorbance of cobalt was also examined. The different volumes of acetonitrile in the range 200–600 μL with the addition of 40 μL of C_2_H_4_Cl_2_ were investigated. As shown in Figure 4, the absorbance of extracted phase increased with the increasing volume of acetonitrile up to 500 μL, and then decreased as the volume of acetonitrile increased.

### 3.3. Influence of pH of Test Solution

The separation of metal ions by DLLME involves the formation of a complex, with sufficient hydrophobicity to be extracted into the small volume of sedimented phase. The chelating agent 5-Br-PADMA shows acid-base indicator properties and exists in four species, H_3_ L^3+^, H_2_ L^2+^, HL^+^ and L, in the solution, with protonation of the ring nitrogen and the two amino group nitrogen atoms of the molecule. Hence, pH plays a unique role on the metal-chelate formation and subsequent extraction. The effect of pH on absorbance was investigated in the range of 4.0–9.0 using Britton–Robinson buffer solution. As shown in Figure 5, the highest absorbance was obtained at pH 6. Thus, a pH of 6 was selected for further study.

### 3.4. Effect of the 5-Br-PADAM Concentration

5-Br-PADMA, a laboratory-synthesized chelating agent, was employed to form a complex with cobalt. The effect influence of the amount of 5-Br-PADMA on the analytical responses was studied. As can be seen in Figure 6, the absorbance signal of Co was firstly increased with increasing amounts of 5-Br-PADMA, and then remained stable in the range of 0.03–0.08 μM (60~160 μL) of 5-Br-PADMA. However, the signal decreased gradually when the amount of 5-Br-PADMA was above 0.08 μM (160 μL). This is because 5-Br-PADMA also had strong hydrophobicity, and there was more 5-Br-PADMA and less complex in the organic phase with the further increase in amounts of 5-Br-PADMA. In this study, 0.06 μM (120 μL) of 5 × 10^−4^ mol·L^−1^ 5-Br-PADMA solution was chosen.

### 3.5. Effect of Extraction Time

The extraction time in this method is defined as the time interval between the moment of injection of disperser solvent acetonitrile, and that of starting the centrifugation process. The effect of extraction time was investigated in the range of 20–180 s, with constant experimental conditions. It was found that the absorbance signal of Co remained constant after the extraction time exceeded 120 s. Thus, an extraction time of 120 s was employed for all measurements.

### 3.6. Effect of Centrifugation Time

Centrifugation time is a significant parameter that affects the separation of the extraction solvent from the aqueous phase. A centrifugation time was studied in the range of 3–8 min at a rate of 4000 rpm. The obtained results showed that the two immiscible phases could be completely separated when the centrifugation time is over 3 min. Hence, a 5 min centrifugation time at 4000 rpm was used.

### 3.7. Effect of Salt

In traditional liquid–liquid extraction, the addition of salt decreases the solubility of analyte in the aqueous solution and enhances its partitioning into the organic phase. Hence, the influence of salt addition on the performance of DLLME was evaluated at 0–5% (*w*/*v*) NaCl levels in this work. The results showed that salt addition had no significant effect on the extraction efficiency of cobalt, which suggest the possibility of using this method for separation of cobalt from saline solutions such as sea water.

### 3.8. Interferences

The effect of coexisting ions in the extraction of cobalt was investigated under optimized conditions. The experiment was performed by analyzing 5.0 mL of standard solutions containing 0.5 ng/mL of cobalt and adding different amounts of potentially interfering ions. The tolerance limit of each concomitant ion was taken as the largest amount causing an error in the determination of cobalt not exceeding 5%. The results showed that 10,000-fold amounts of Na^+^, K^+^, Mg^2+^, and Ca^2+^; 2000-fold amounts of Sr^2+^; 1000-fold amounts of Ba^2+^, Cl^−^, and NO_3_^−^; 800-fold amounts of Pb^2+^, Cd^2+^, Hg^2+^, La^3+^, Fe^3+^, Al^3+^, Cr^3+^, As (III), F^−^, and SO_4_^2−^; 500-fold amounts of Mn^2+^ and Mo (VI); 100-fold amounts of Ni^2+^, Cu^2+^, and I^−^; 70-fold amounts of Bi (III) and Sn (IV); and 50-fold amounts of Ce (IV) did not interfere with the measurements. These results clearly demonstrated the high selectivity of the developed D-DLLME method for the determination of trace cobalt in environmental samples.

### 3.9. Analytical Figures of Merit Calibration Curve, Detection Limit and Precision

Under the optimized experimental conditions for the determination of cobalt with the proposed DLLME-GFAAS method, the calibration curve was linear over the range of 0.05–1.0 ng/mL cobalt. The linear equation was *A* = 0.0222 + 1.0356 *c*, where *A* is the absorbance, *c* is the cobalt concentration in solution (ng/mL), and the correlation coefficient was R = 0.9922. The detection limit (DL), calculated as three times the standard deviation of the blank, was 0.02 ng/mL (3*σ*). The relative standard deviation (RSD) for six replicate measurements of 0.5 ng/mL cobalt was 3.8%. In order to determine the preconcentration factor, an analytical graph was prepared without the preconcentration step for cobalt in the range of 20–100 ng/mL. The calibration equation obtained was *A* = 0.0759 + 0.009204 *c*. Preconcentration factor (PF), calculated as the ratio of the aqueous solution volume (5 mL) to that of the organic phase volume (40 μL), and enrichment factor (EF), calculated as the ratio of the slopes of the calibration graphs obtained with and without DLLME for 5.0 mL sample solution, were 120 and 112.5, respectively. The extraction efficiency of the proposed method calculated by (EF/PF) × 100) was 93.8%.

Table 2 compares the characteristic data of the developed method, with the previously reported methods concerned with cobalt preconcentration by DLLME in terms of detection limits and instruments employed.

### 3.10. Analytical Applications

The proposed method was successfully applied for the preconcentration and speciation of trace amounts of cobalt in real water samples, and the results are given in Table 3. As can be seen, the recoveries for the spiked samples were within the range of 96.5–103.0%, indicating that the present method is reliable and suitable for the determination of cobalt in water samples.

## 4. Conclusions

In this study, we proposed a novel method for the determination of ultra-trace levels of cobalt by DLLME, combined with GFAAS. This method is based on the complexation of cobalt(II), with a laboratory-synthesized reagent 5-Br-PADMA, to form a complex that can be extracted into fine droplets of 1,2-dichloroethane (CH_2_ClCH_2_Cl) using acetonitrile (CH_3_CN) as a dispersive solvent. The method offers several advantages such as simplicity, low cost, lower limit of detection, and a higher enrichment factor. In particular, the consumption of toxic organic solvents is minimized, meeting the requirements of green chemistry. The developed method has been successfully applied to the determination of cobalt in river, reservoir, and well water samples, and the precision and accuracy of the method are satisfactory. The results show that this method is a sensitive, efficient, and rapid sample preparation technique for different real samples and can be used as a replacement of the traditional extraction method for determination of cobalt.

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
