# Peer review of "Determination of Ultra-Trace Cobalt in Water Samples Using Dispersive Liquid-Liquid Microextraction Followed by Graphite Furnace Atomic Absorption Spectrometry"

_molecules, 2022, doi:10.3390/molecules27092694_

Round 1

Reviewer 1 Report

 The work under consideration is devoted to the problem of detecting trace amounts of cobalt in water. Cobalt compounds are widely used in various branches of industry. However, cobalt pollution can lead to negative (toxic) consequences for living organisms. Therefore, the determination of trace amounts of cobalt in aqueous solutions is an urgent task. 

To date, there are a number of methods that allow one to analyze low concentrations of cobalt in aqueous media (the authors provide references to these methods in the Introduction). The advantage and novelty of this work are the combination of two previously known methods - dispersive liquid-liquid microextraction (DLLME) and graphite furnace atomic absorption spectrometry (GFAAS). The DLLME method is highly efficient in preconcentration of Co in a very small volume of solvent. The authors used this method by forming a Co(II) complex with a laboratory-synthesized reagent named 5-Br-PADMA and consequent extracting it from the solution. In turn, GFAAS is a highly sensitive detection technique and requires only a few microliters of the sample to carry out the determination. The combination of these methods leads to the possibility of determining cobalt with high sensitivity and accuracy with the use of very small volumes of reagents (an enrichment factor is 210, a limit of detection is 0.03 ng/mL). In addition, the method described in this work is quite simple and has a low cost, which may lead to its widespread use.    

            The authors investigated the effect of a number of parameters on the extraction efficiency. The dependencies on types and volumes of extraction and disperser solvents, pH of sample solution, extraction time, concentration of the chelating agent 5-Br-PADMA, and salt effect were studied, and optimal conditions for analysis were selected.

I have no negative comments on the content of the article and the form of its presentation. One note is: The heading under Fig. 6 (line 200) should be corrected - it should be "Effect of the 5-Br-PADAM concentration..." instead of "Effect of pH of the sample solution..."

In general, this article contains novelty; the method described in the article may be useful in determination of trace amount of cobalt in water solutions. Thus, the article can be published taking into account the above remark.

Author Response

Dear reviewer 1:

Thank you very much for your comments. 

For your guidance, responses to you are appended below.

With the best regards

Yours sincerely,

Han Quan

School of Chemical Engineering, Xi′an University, Xi′an, Shaanxi Province, China.

April 8, 2022

-------------------------------------------------------------------------------------------------------

The following is a point-to-point response to your comments.

To Reviewer #1:

Question 1: I have no negative comments on the content of the article and the form of its presentation. One note is: The heading under Fig. 6 (line 200) should be corrected - it should be "Effect of the 5-Br-PADAM concentration..." instead of "Effect of pH of the sample solution..."

Answer: Sorry for the mistake. Corrected, marked in red color in the revised manuscript (line 200).

Reviewer 2 Report

The manuscript presented a new method for the determination of ultra-trace levels of cobalt by DLLME combined with GFAAS. Authors used  a laboratory-synthesized reagent 5-Br-PADMA to form a complexes  of cobalt(ІІ) which can be extracted into the fine droplets of 1,2-dichloroethane using acetonitrile as dispersive solvent. The method was applied to the determination of cobalt in river, reservoir and well water samples, and the precision and accuracy of the method are satisfactory. The manuscript demonstrated a complete optimization of DLLME extraction parameters and analytical performances. These are some comments that should be revised before publication.

  1. Abstract:  Line 12  Use full name of "acetate buffer" instead of using the abbreviation of  "HAc-NaAc".
  2. The extraction efficiency of the proposed method should be included in the manuscript (Calculated by =(EF/PF)*100 ) .
  3. What is preconcentration factor (PF) of the proposed method which is defined as Sample volume/final extract volume )?
  4. General procedure:  Line 108 "1.0 mL pH 6.0 Britton-Robinson buffer solution"  In the abstract, acetate buffer was used to adjust pH. Please identify certain type of buffer used in the proposed method. 
  5. Since authors used a new laboratory-synthesized reagent 5-Br-PADMA, a brief synthesis procedure could be included as referred to reference number 31.
  6. The chemical modifier used in GFAAS measurement  should be included in "Reagents and solution". Author should describe type of chemical modifier , concentration and volume used for GFAAS analysis.
  7. Figure 1:  Use "solvent type" Line 141
  8. Figure 6: Convert volume of 5-Br-PADAM in  µL to Concentration of 5-Br-PADAM in mM on X axis.
  9. For method validation, the certified reference material (CRM) / standard reference material (SRM) / quality control sample (QC samples) may be necessary to validate the proposed method and statistical test i.e. t-test may be required. 
  10. Conclusions: Line 287  Please use the determination of cobalt in
    river, reservoir and well water samples instead of "the speciation".  Use small letter for "reservoir" Line 288 . Use "a laboratory-synthesized reagent 5-Br-PADMA" line 282.

Author Response

Dear reviewer 2:

We would like to thank you very much for giving us constructive suggestions which would help us both in English and in depth to improve the quality of the paper. For your guidance, responses to you are appended below.

We would appreciate your in-depth review and suggestions for my submission.

With the best regards

Yours sincerely,

Han Quan

College of Chemical Engineering,  Xi′an University,  Xi′an,Shaanxi Province China.

April 8, 2022

-------------------------------------------------------------------------------------------------------

The following is a point-to-point response to your comments.

To Reviewer #2:

Question 1: Abstract:  Line 12  Use full name of "acetate buffer" instead of using the abbreviation of  "HAc-NaAc".

Answer: Sorry for making the mistake. It should be Britton-Robinson buffer solution. Corrected in the resubmitted manuscript in red color.

The abbreviation of "HAc-NaAc" is replaced by “acetic acid-sodium acetate” in the resubmitted manuscript. See line 12 in red color.

Question 2: The extraction efficiency of the proposed method should be included in the manuscript (Calculated by =(EF/PF)∙100 ) .

Answer: Thank you for your suggestion. The extraction efficiency (Calculated by =EF/PF*100) of the proposed method has been given in the resubmitted manuscript. See line 22 in red color.

Question 3: What is preconcentration factor (PF) of the proposed method which is defined as Sample volume/final extract volume )?

Answer: Yes, the definition of “preconcentration factor (PF)” has been given in the resubmitted manuscript. See line 19 in red color.

Question 4: General procedure:  Line 108 "1.0 mL pH 6.0 Britton-Robinson buffer solution"  In the abstract, acetate buffer was used to adjust pH. Please identify certain type of buffer used in the proposed method.

Answer: Sorry for making the mistake. It should be Britton-Robinson buffer solution. Corrected in the resubmitted manuscript in red color.

Question 5: Since authors used a new laboratory-synthesized reagent 5-Br-PADMA, a brief synthesis procedure could be included as referred to reference number 31.

Answer: Thank you for your suggestion. The synthesis procedure is given in the resubmitted manuscript. See page 3 of 12, in red color.

Question 6: The chemical modifier used in GFAAS measurement should be included in "Reagents and solution". Author should describe type of chemical modifier, concentration and volume used for GFAAS analysis.

Answer: Added in the resubmitted manuscript. See page 3 of 12, in red color.

Question 7: Figure 1:  Use "solvent type" Line 141

Answer: Revised accordingly.

Question 8: Figure 6: Convert volume of 5-Br-PADAM in µL to Concentration of 5-Br-PADAM in mM on X axis.

Answer: Revised accordingly. See page 7of 12, in red color.

Question 9: For method validation, the certified reference material (CRM) / standard reference material (SRM) / quality control sample (QC samples) may be necessary to validate the proposed method and statistical test i.e. t-test may be required.

Answer: Thank you for your suggestion. But because of lacking certified reference material (CRM) / standard reference material (SRM), we do not revise accordingly. Hope you to understand.

Question 10: Conclusions: Line 287.  Please use the determination of cobalt in river, reservoir and well water samples instead of "the speciation".  Use small letter for "reservoir" Line 288 . Use "a laboratory-synthesized reagent 5-Br-PADMA" line 282.

Answer: Sorry for making the mistake. Revised. See page 10of 12, in red color.

Reviewer 3 Report

The manuscript is well-conducted and its of interest to the readers. However, there are some minor aspects that should be addressed.

  • Material and methods: please, add the information in bracket about the brand and origin country of the reagents and laboratory material used.
  • Results&Discussion: Could you compare your results with other methods used to determine Cobalt?
  • Conclusions: what is about future concerns of the method ?
  • References: some references are not in the journal format. Revise.

Author Response

Dear reviewer 3:

We would like to thank you very much for giving us constructive suggestions which would help us to improve the quality of the paper. For your guidance, responses to you are appended below.

We would appreciate your in-depth review and suggestions for my submission.

With the best regards

Yours sincerely,

Han Quan

College of Chemical Engineering, Xi′an University, Xi′an China.

Aprir 8, 2022

-------------------------------------------------------------------------------------------------------

The following is a point-to-point response to your comments.

To Reviewer #3:

Question 1: The manuscript is well-conducted and its of interest to the readers. However, there are some minor aspects that should be addressed. Material and methods: please, add the information in bracket about the brand and origin country of the reagents and laboratory material used.

Answer: Added accordingly.

Question 2: Results&Discussion: Could you compare your results with other methods used to determine Cobalt?

Answer: Thank you very much for your suggestion. Because it is not convenient for us to find a suitable method to determine cobalt and make comparison with the proposed method under present conditions and it will take a relatively longer time, we do not revise accordingly. Hope you to understand.

Question 3: Conclusions: what is about future concerns of the method ?

Answer: Added in the resubmitted manuscript. See page 10 of 12, in red color.

Question 4: References: some references are not in the journal format. Revise.

Answer: Revised accordingly.